# Multi-modal deep learning framework for damage detection in social media posts

Jiale Zhang[1], Manyu Liao[1], Yanping Wang[1], Yifan Huang[2], Fuyu Chen[3] and Chiba Makiko[4]

[1] School of Journalism and Communication, Nanchang University, Nanchang, China
[2] School of International Relations and Diplomacy, Beijing Foreign Studies University, Beijing, China
[3] Journalism and Information Communication School, Huazhong University of Science and Technology, Wuhan, China
[4] School of Foreign Languages, Zhejiang University of Technology, Zhejiang, China

## ABSTRACT

In crisis management, quickly identifying and helping affected individuals is key, especially when there is limited information about the survivors' conditions. Traditional emergency systems often face issues with reachability and handling large volumes of requests. Social media has become crucial in disaster response, providing important information and aiding in rescues when standard communication systems fail. Due to the large amount of data generated on social media during emergencies, there is a need for automated systems to process this information effectively and help improve emergency responses, potentially saving lives. Therefore, accurately understanding visual scenes and their meanings is important for identifying damage and obtaining useful information. Our research introduces a framework for detecting damage in social media posts, combining the Bidirectional Encoder Representations from Transformers (BERT) architecture with advanced convolutional processing. This framework includes a BERT-based network for analyzing text and multiple convolutional neural network blocks for processing images. The results show that this combination is very effective, outperforming existing methods in accuracy, recall, and F1 score. In the future, this method could be enhanced by including more types of information, such as human voices or background sounds, to improve its prediction efficiency.

# INTRODUCTION

## Background

In crisis management, swiftly detecting and assisting affected individuals are primary concerns for emergency responders (*Barton, 1994*). Limited or interrupted information about survivors' conditions exacerbates this problem (*Saroj & Pal, 2020*). Conventional emergency communication systems frequently encounter constraints in terms of their ability to be reached and their capacity, particularly during periods of high demand. This limits the effectiveness of response efforts (*Saroj & Pal, 2020*; *Wut, Xu & Wong, 2021*; *Ritchie & Jiang, 2021*). Over the years, social media platforms have become critical adjunct resources, particularly in disasters (*Phengsuwan et al., 2021*; *Muhammed & Mathew, 2022*).



Corresponding author
Chiba Makiko,
makiko_zjut@outlook.jp

During Hurricane Harvey, a Category 4 hurricane that hit Texas and Louisiana in August 2017, over 2.46 billion communications, such as calls and messages, were sent to report locations, conditions, and other vital information (*Mirbabaie et al., 2020*). During Hurricane Sandy, a massive and destructive Category 3 Atlantic hurricane that devastated the Caribbean and the coastal Mid-Atlantic region of the United States in late October 2012, approximately 25 first aid officers were tasked with managing around 2.5 million disaster-related social media posts (*Meng & Mozumder, 2021*). A woman received assistance after posting requests for help on Twitter when her 911 calls went unanswered (*Zou et al., 2023*). Similarly, during Hurricane Harvey, social media was crucial in rescuing individuals unreachable through conventional emergency call systems (*Zou et al., 2023*). These examples demonstrate the essential role of social media in facilitating immediate and effective responses in rescue activities (*Mirbabaie et al., 2020*; *Yuan et al., 2021*; *Zou et al., 2023*). Nonetheless, the immense amount of data obtained *via* social media during emergencies poses significant difficulties. Engaging in the manual processing of large amounts of information is not feasible and takes emergency responders away from important duties, which might negatively impact overall crisis management efforts (*Zhang, Ding & Ma, 2020*). Consequently, there is an imperative need for the development and implementation of automated systems capable of efficiently parsing and extracting actionable intelligence from the deluge of social media data (*Fan, Wu & Mostafavi, 2020*; *Imran et al., 2020*). Such technological advances would significantly enhance the operational capabilities of emergency response teams, allowing for more strategic allocation of resources and improving the timeliness and effectiveness of crisis interventions (*Sufi & Khalil, 2022*). Leveraging automated analytical tools to navigate and utilize social media data represents a crucial evolution in emergency management methodologies, aiming to optimize response actions and ultimately save lives (*Sufi & Khalil, 2022*).

## Related works

The humanitarian computing community is setting its sights on the creation of automated systems capable of identifying and marking social media posts that provide data on disasters and crises (*Coppi, Moreno Jimenez & Kyriazi, 2021*; *Sufi & Khalil, 2022*). These systems are structured into three primary components. The first component focuses on retrieving social media posts that may be relevant to a disaster or crisis (*Kaufhold, Bayer & Reuter, 2020*). This stage is crucial as it involves sifting through a vast array of irrelevant posts, isolating those that are genuine and timely, among other vital criteria (*Kaufhold, Bayer & Reuter, 2020*). The second component deals with the categorization of these filtered posts, distinguishing between different types of damage and disaster incidents and ensuring that each post is evaluated for its value in conveying substantive information about the situation at hand (*Asif et al., 2021*; *Wang et al., 2023*). The final component is concerned with condensing the information in these posts to distill essential disaster-related data, which is then formatted to be quickly and effectively communicated to emergency response personnel (*Asif et al., 2021*). By processing and analyzing social media content through these modules, the automated systems aim to enhance the situational

awareness of emergency responders. This advancement in humanitarian computing not only streamlines the flow of crucial information in times of crisis but also ensures a more agile and effective response to disasters, ultimately aiding in saving lives and reducing damages (*Coppi, Moreno Jimenez & Kyriazi, 2021*; *Sufi & Khalil, 2022*). Various applications in humanitarian computing have utilized social media posts, spanning event detection (*Sakaki, Okazaki & Matsuo, 2013*; *Yang et al., 2024*), the development of alert systems (*Breen, Ida & Vidhya, 2016*), map creation (*Cresci et al., 2015*), and the extraction of actionable intelligence (*Caragea et al., 2011*; *Ashktorab et al., 2014*). A crucial aspect of these applications is identifying the type and extent of damages to infrastructures, natural environments, or individuals, which aids first responders in effectively deploying resources. Research in damage detection has explored different approaches, including text analysis (*Imran et al., 2014*; *Cresci et al., 2015*; *Imran, Mitra & Castillo, 2016*; *Nguyen et al., 2016*; *Dang et al., 2023*), image processing (*Alam, Imran & Ofli, 2017*; *Nguyen et al., 2017*), or combinations of both approaches (*Jomaa, Rizk & Awad, 2016*). *Imran et al. (2014)* focused on text-based tweet analysis to identify damage information, while *Cresci et al. (2015)* applied natural language processing to tweets for damage assessment. *Nguyen et al. (2016)* employed artificial neural networks (ANN) to categorize tweets according to their relevance to damages. In terms of the image analysis, *Alam, Imran & Ofli (2017)* implemented a deep learning framework to evaluate damage type and severity in images. *Nguyen et al. (2017)* enhanced convolutional neural networks (CNN) using domain-specific images to improve the efficiency of damage detection from visual data. Besides, several studies have integrated text and image analysis into a multimodal approach. *Jomaa, Rizk & Awad (2016)* combined basic visual feature extraction with textual analysis to classify damage in tweeted images, showing superior results compared to single-mode classifiers.

Understanding visual scenes and semantics is vital for damage identification and extracting actionable insights. Deep learning stands out in the field for its ability to comprehend scenes and semantics from extensive and often unstructured data sets, requiring minimal manual feature extraction (*LeCun, Bengio & Hinton, 2015*). CNN was initially created for visual understanding (*Krizhevsky, Sutskever & Hinton, 2017*) and later adapted for analyzing text (*Kim, 2014*) and sound (*Abdel-Hamid et al., 2014*). CNNs, trained on both labeled and unlabeled data, excel at recognizing a broad spectrum of objects, even achieving accuracies beyond human capabilities (*Krizhevsky, Sutskever & Hinton, 2017*). These networks process input data, like image pixels, through layered convolutions and nonlinear pooling to identify various categories, as evidenced by their performance on the ImageNet challenge, outperforming other methods by nearly 11% with an error rate of 15% (*Krizhevsky, Sutskever & Hinton, 2017*). Inception, a deep 15-layer CNN, has set new benchmarks in image recognition (*Szegedy et al., 2017*). Recurrent neural network (RNN), particularly long short-term memory (LSTM), have also shown impressive results in visual scene analysis, notably in generating accurate image captions (*Lai et al., 2015*). Regarding semantic understanding, deep learning has been pivotal in developing word embeddings, representing words as dense vectors (*Mikolov et al., 2013*). This approach supports deeper sentence and textual comprehension,

facilitated by word embedding techniques like Word2Ve (*Mikolov et al., 2013*). Deep learning has led to breakthroughs in sentiment analysis (*Glorot, Bordes & Bengio, 2011*), sentence classification (*Lai et al., 2015*), text generation (*Vinyals et al., 2015*), and other linguistic tasks. Additionally, CNNs have significantly contributed to deep learning's achievements in the domains of text classification (*Kim, 2014*; *Zhang, Zhao & LeCun, 2015*; *Conneau et al., 2017*), sentiment analysis (*dos Santos & Gatti, 2014*), and the acquisition of semantic knowledge (*Shen et al., 2014*), showcasing their versatility and effectiveness across different domains of data processing.

In order to fully use the capabilities of data originating from numerous sources with distinct characteristics (such as size, structure, *etc.*), it is necessary to develop machine learning algorithms that can effectively combine and utilize these diverse data types. Multimodal learning addresses this by evolving or tailoring machine learning algorithms to assimilate information from various modalities (*Ramachandram & Taylor, 2017*). *Guillaumin, Verbeek & Schmid (2010)* enhanced image categorization by integrating tags into the feature vectors, training a multiple kernel learning classifier, and achieving an improvement of nearly 10% in accuracy for certain categories. *Alqhtani, Luo & Regan (2015)* improved event detection from Twitter data by incorporating semantic and visual features to train a *k*-nearest neighbor model with an 8% increase in classification accuracy over single-mode methods. *Jomaa, Rizk & Awad (2016)* utilized a method combining visual and semantic features in feature vectors, employing support vector machines to refine damage classification, resulting in an accuracy enhancement of 4%. *Poria et al. (2016)* conducted sentiment analysis using a mix of text, audio, and image data, applying both feature fusion and deep fusion methods, surpassing the precision of leading-edge techniques by at least 20%. While these studies separated feature extraction and supervised learning, there has been a move toward deep learning multimodal algorithms in unsupervised learning settings (*Ngiam et al., 2011*). *Ngiam et al. (2011)* developed an ANN model to learn from raw audio and visual data. *Srivastava & Salakhutdinov (2012)* employed deep Boltzmann machines for learning from image and text data.

## Proposed method

In our research, we introduce a computational framework designed for detecting damage in social media posts, employing the Bidirectional Encoder Representations from Transformers (BERT) architecture (*Devlin et al., 2018*) alongside a multi-block convolutional approach. Our approach consists of two key elements: a network grounded in BERT for extracting text features, and CNN blocks dedicated to image feature extraction. We expect this integration to yield a performance that surpasses existing methods. By leveraging the strengths of both text and visual data processing, our method aims to provide a comprehensive and nuanced analysis of social media content, enhancing the accuracy and reliability of damage detection in real-world scenarios. This synergistic approach harnesses the deep learning capabilities of BERT and CNN to effectively interpret the complex and varied data found in social media posts, potentially setting a new standard for automated damage assessment tools. In this work, we highlighted our

contribution to designing a streamlined architecture for the image processing stage, instead of using pre-trained models like other approaches. The introduction of streamlined processing significantly enhances the training speed as well as adaptability to specific datasets, especially small ones.

# METHOD

Our bimodal deep learning model, which utilizes both visual and textual data, can enhance the performance of damage detection tasks by leveraging complementary information. Visual data provides detailed physical characteristics of the damage, while textual data offers contextual details and descriptions. This integration allows for improved accuracy and robustness, as the model can cross-validate information and extract diverse features. Additionally, textual data helps disambiguate visual ambiguities and incorporates expert knowledge, leading to a more comprehensive and adaptive understanding of various damage scenarios. Consequently, bimodal models can more effectively identify and classify damage, improving overall predictive performance.

## Proposed architecture

To address the problem, we proposed a model that combines both text and image features to enhance prediction efficiency. The proposed model consists of two components: text processing (TP) and image processing (IP) (Fig. 1). In the TP component, a pre-trained BERT model was used to extract text features. For the IP component, we developed a network comprising three convolutional (Conv) blocks, one Transformer block, one 2D-convolutional (Conv2D) layer, and one fully connected (FC) layer. Each sample is represented by a pair of image and text. The images, of size $224 \times 224 \times 3$, are first passed through a Conv2D layer before entering the IP component. The Conv block is designed with two Conv2D layers, with a batch normalization (BatchNorm) layer following each Conv2D layer, and rectified linear unit (ReLU) is used as the activation function. The vectors of image features and text features are then concatenated before passing through the last FC layer. The Softmax function is employed to return predictions.

## BERT-based pretrained model

The BERT architecture (*Devlin et al., 2018*), initially introduced by Google, has been a groundbreaking development in the natural language processing (NLP) domain for text information extraction. The core innovation of BERT is its bidirectional processing capability using Transformers, an attention mechanism that comprehends the contextual relationship of words or sub-words within the text. In contrast to traditional models that process text linearly, either from left to right or right to left, BERT analyzes the entire text string simultaneously. This feature enables it to grasp the context on both sides, significantly enhancing its ability to understand each word's meaning within its contextual environment. BERT-based pretrained model was developed using a large *corpus* of text from the Internet, including the entire Wikipedia (in English), which is

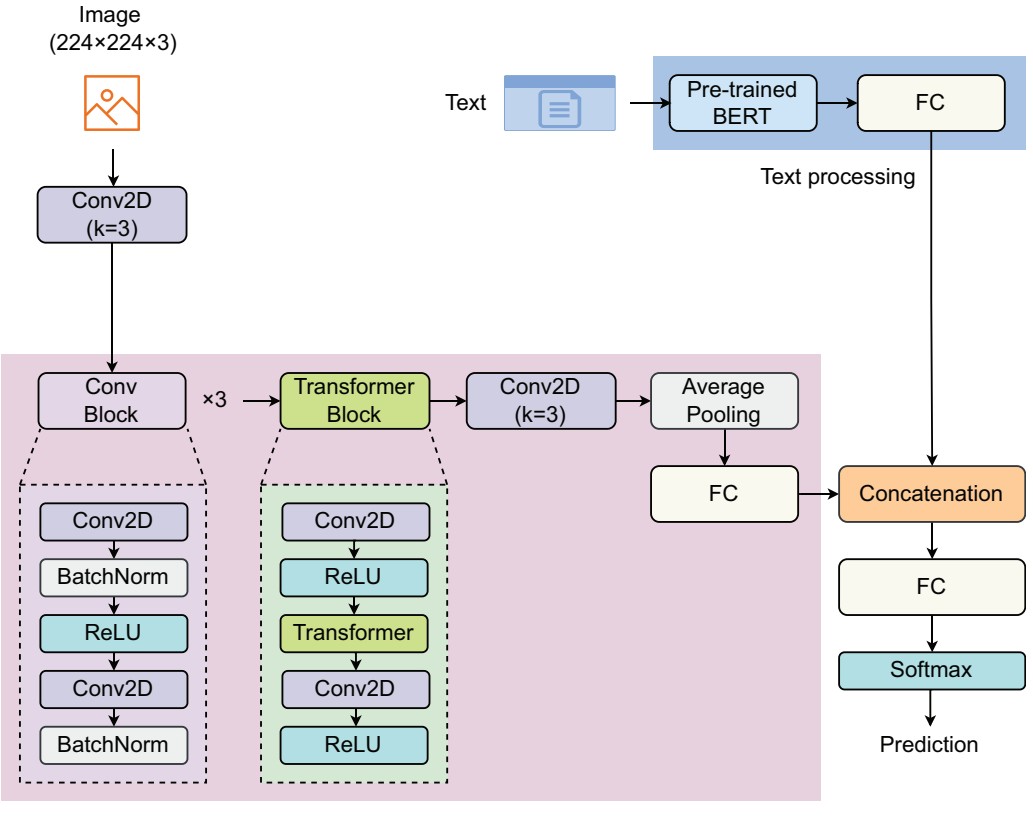

**Figure 1 Proposed model for detecting social media posts.** The model contain two main components: image processing and text processing.

2,500 million words long. During its training, BERT uses two main strategies: masked language model (LLM) and next sentence prediction (NSP). While LLM randomly masks 15% of the words in a sentence and then predicts the masked words based on the context provided by the other non-masked words in the sentence, NSP also learns to predict whether two given segments of text naturally follow each other, which helps it understand the relationship between consecutive sentences. After the pretraining, BERT can be fine-tuned with just one additional output layer for various NLP tasks without substantial task-specific architecture modifications. For textual information extraction, BERT evaluates the context of words and sentences in the text thoroughly, which enables it to extract entities, relationships, sentiments, or any other information effectively. This ability to process words in relation to all other words in a sentence allows BERT to establish a nuanced understanding of language nuances and complexities, leading to more accurate extraction of textual information for tasks such as sentiment analysis, entity recognition, summarization, and question answering. The maximum length of the text is set to 300. The input text and attention mask have a size of $1 \times 300$. After passing through the BERT-based pretrained model, the output vector with a size of $1 \times 768$ is returned, which is then passed through the next linear layer.

## Convolutional block

Before entering the convolutional (Conv) block, the input image ($X_{input}$) with a size of $224 \times 224 \times 3$ enters the first Conv2D layer (Conv2D), specified by a kernel size of $3 \times 3$, to create a corresponding feature map ($X_{conv}$) with a size of $112 \times 112 \times 16$. This feature map is the input of these next Conv blocks. Our Conv block, with its sequence of layers, effectively enhances the network's capability to extract and learn robust feature representations. Each Conv block two Conv layers termed as $Conv2D_1$ and $Conv2D_2$, respectively. The activated feature map ($X_{activated\_conv}$) is then consecutively passed through three Conv blocks, represented mathematically as:

$$X_{norm1} = \text{BatchNorm}(X_{conv1}) = \text{BatchNorm}(Conv2D_1(X_{activated\_conv})), \qquad (1)$$

where $X_{norm1}$ is the normalized feature map obtained after the feature map ($X_{conv1}$) are passed through the BatchNorm layer. The BatchNorm layer standardizes the activations from the previous layer, improving training stability and speed. After normalization, the ReLU activation function is applied on, introducing non-linearity and allowing the network to learn complex patterns.

Next, the activated output passes through another Conv2D layer ($Conv2D_2$), further processing the features to create the secondary feature map ($X_{conv2}$) with the same size as primary feature map ($X_{conv1}$):

$$X_{conv2} = Conv2D_2(\text{ReLU}(X_{norm1})). \qquad (2)$$

Finally, the secondary feature map $X_{conv2}$ is normalized in another BatchNorm layer and complete the block:

$$X_{conv\_out\_i} = \text{BatchNorm}(X_{conv2}), \qquad (3)$$

where $X_{conv\_out\_i}$ (i = {1, 2, 3}) is the final output of the Conv block. The sizes of the convoluted outputs $X_{conv\_out\_i}$ are gradually reduced from $112 \times 12$ to $56 \times 56$, and finally to $28 \times 28$ while the output's depth increases from 16 to 32.

## Transformer block

The Transformer block is developed by a sequence of layers designed to process input data effectively. The input of the Transformer block is the output feature map of the third Conv block ($X_{conv\_out\_3}$). Initially, the input passes through a 2D convolutional layer ($Conv2D_{T1}$), mathematically represented as:

$$H_{convT1} = \text{ReLU}(Conv2D_T(X_{conv\_out\_3})), \qquad (4)$$

where $H_{convT1}$ is the output feature map. After passing the first Conv2D layer, the output $H_{convT1}$ is fed into the Transformer layer. The core of the Transformer layer is the self-attention mechanism, which can be formulated as:

$$\text{Attention}(Q, K, V) = \text{softmax}\left(\frac{QK^T}{\sqrt{d_k}}\right)V. \qquad (5)$$

In this formula, $Q$, $K$, and $V$ represent the queries, keys, and values matrices, respectively, derived from $\mathbf{X}_{conv1}$, and $d_k$ is the dimensionality of the keys. The output from the Transformer layer, denoted as $\mathbf{H}_{trans}$, is then processed through another Conv2D layer (Conv2D$_{T2}$), followed by a ReLU activation function, yielding the final output $\mathbf{H}_{output}$ with the same size as the input. This process can be described mathematically as:

$$\mathbf{H}_{output} = \mathrm{ReLU}(\mathrm{Conv2D}_{T2}(\mathbf{H}_{trans})). \qquad (6)$$

Our Transformer block, characterized by its integration of Conv layers and a Transformer layer, is strategically engineered to harness both local and global dependencies within the dataset. This dual-layer architecture ensures a comprehensive analysis, where the Conv layers focus on extracting spatial features by recognizing patterns and textures in the data, while the Transformer layer excels in capturing the global context, enabling the model to understand broader relationships and dependencies. This synergy enhances the model's ability to discern nuanced features and relationships that are crucial for complex data interpretation tasks. The Conv layers, with their inherent strength in dealing with structured data, process the input in a hierarchical manner, identifying local features at various levels of abstraction. The Transformer layer, on the other hand, employs attention mechanisms to weigh and integrate these features across the entire input space, thus providing a dynamic and context-aware processing capability. This combination not only improves the accuracy of feature extraction but also increases the robustness of the model, making it adaptable to a wide range of scenarios. By effectively capturing and integrating both the detailed and overarching elements of the data, our Transformer block is suitable for advanced analytical tasks. It paves the way for more sophisticated processing strategies, potentially leading to breakthroughs in various applications such as image recognition (*Zhou et al., 2024*; *Zhao et al., 2021*; *Wang et al., 2021*; *Hu et al., 2021*), NLP (*Wolf et al., 2020b*, *2020a*; *Kalyan, Rajasekharan & Sangeetha, 2021*), and predictive analytics (*Kabir, Foggo & Yu, 2018*; *Bukhsh, Saeed & Dijkman, 2021*; *Park et al., 2023*). Comprehending the complex interaction between local and global data characteristics is essential for obtaining the most effective outcomes.

## DATASET

The dataset used in this study was collected from *Mouzannar, Rizk & Awad (2018)*. A collection of 5,879 images with captions from social media depicting damage caused by natural disasters or wars is categorized into six classes: *Fires*, *Floods*, *Natural Landscapes*, *Infrastructure*, *Human*, and *Non-damage*. According to *Mouzannar, Rizk & Awad (2018)*, the refined dataset was processed *via* multiple steps. Social media platforms (*e.g.*, Instagram, Twitter, and Facebook) often serve as initial sources for reporting emergency and crisis events and providing crucial information. However, the challenge lies in the overwhelming presence of irrelevant or uninformative content, complicating the extraction of useful information for emergency response teams (*Olteanu, Vieweg & Castillo, 2015*). Efforts have been made to filter disaster-specific content on Twitter, analyzing both the textual tweets (*Abel et al., 2012*; *Chowdhury et al., 2013*; *Olteanu,*

*Vieweg & Castillo, 2015*; *Jomaa, Rizk & Awad, 2016*) and their associated images (*Jomaa, Rizk & Awad, 2016*; *Alam, Imran & Ofli, 2017*). For a sample, it is represented by a pair of image and text (Fig. 2).

This dataset was constructed with a particular focus on Instagram as it is inherently suited for the scope of the study, and since these posts primarily exist as captioned images, the process of correlating text with visual content is simplified. To ensure a comprehensive representation of crisis-related images and captions, data were gathered using more than 100 hashtags across various time frames. Additionally, the textual data was enriched by incorporating informative tweets from the CrisisLexT26 dataset (*Olteanu, Vieweg & Castillo, 2015*) and tweets related to human and infrastructure damage from the CrisisNLP dataset (*Imran, Mitra & Castillo, 2016*). The image dataset was expanded by including images obtained from Google searches using the same keywords and hashtags. To remove captioned images that were not relevant, a multimodal decision-based system was employed. The system processed the caption through a text relevancy model and the image through an image relevancy model. When both models concurred on relevancy, the captioned image was then classified into the appropriate category. If there was a discrepancy between the text and image relevancy, they were categorized separately. For a captioned image, if both its text and image matched the respective elements of another post, it was then removed from the dataset. After removing duplicates, the refined dataset was finalized with 5,830 samples. Table 1 gives information on the refined dataset for model training and testing.

## RESULTS AND DISCUSSION

### Model benchmarking

To fairly assess the prediction efficiency of our proposed method against other existing state-of-the-art (SOTA) methods, we compared our model with three others, whose IP components were developed using the AlexNet (*Krizhevsky, Sutskever & Hinton, 2017*), VGG16 (*Conneau et al., 2017*), and ResNet50 (*He et al., 2016*) architectures. These SOTA models being compared have the same architecture in the TP components, while in the IP components, the AlexNet, VGG16, and ResNet50 pre-trained models were used to extract the image features. In addition to SOTA models developed using both image and textual data, we implemented three models processing only image data and three models processing only textual data as baseline models. For the development of image-only models, we still utilized the AlexNet, VGG16, and ResNet50 architectures. To develop text-only models, we used two other architectures: the Cross-lingual Language Model (XLM) (*Lample & Conneau, 2019*) and the Robustly Optimized BERT Pretraining Approach (RoBERTa) (*Liu et al., 2019*) beside the BERT architecture used in our proposed method. The efficacy of our models was assessed using various evaluation metrics. We evaluated other metrics such as accuracy, precision, recall, and F1 scores (F1), all of which were computed using a default threshold of 0.5. Table 2 presents the performance comparisons of these models.

Experimental results indicate that our proposed method achieves superior performance in terms of accuracy, recall, and F1 score. Our model achieves an accuracy of 0.8473,

Text Sample

Here's some video of the smoldering ruins in White Hall, IL. When I left I smelled like a campfire. 3 massive buildings destroyed by a giant fire. A local woman told me about it all. Just unbelievably sad. So much history and such a historic loss in a really small town.

#smalltownusa #whitehall #centralillinois #illinois_shots #architecture #brickstagram #firedamage #buildingruins #exploreillinois #buildingfire

*Hashtag*

Image sample

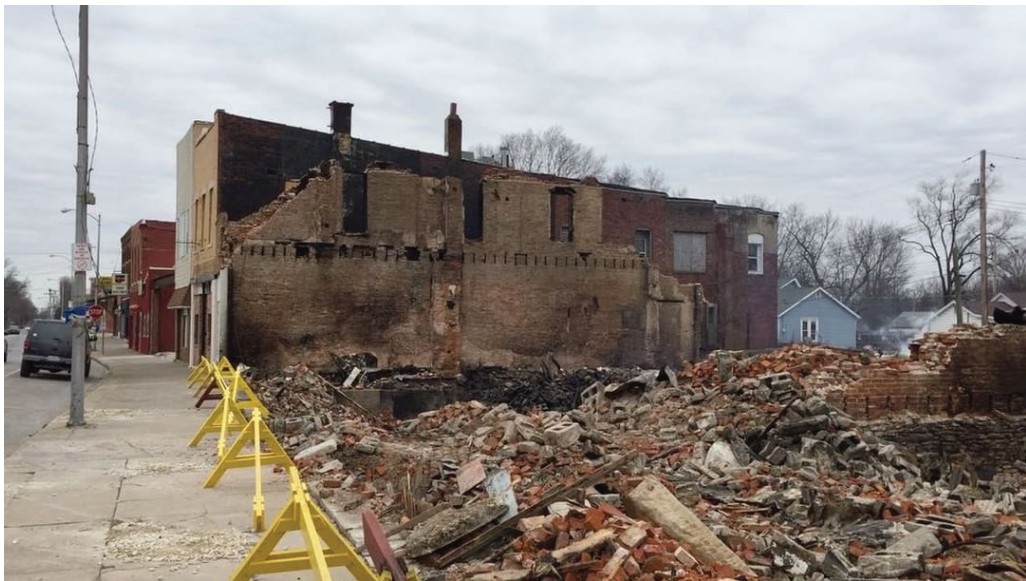

**Figure 2 The text and image sample (ID: buildingfire 2017-02-05 04-06-10).** The dataset is licensed under a Creative Commons Attribution 4.0 International (CC BY 4.0) license and is available at DOI: 10.24432/C52P6P.

outperforming those developed with ResNet50, VGG16, and AlexNet. The precision of the ResNet50-based model is slightly higher than ours but significantly higher than the others. The recall for the VGG16-based model is second only to ours and exceeds both the AlexNet-based and ResNet50-based models. The F1 scores for the VGG16-based and ResNet50-based models are nearly the same, at approximately 0.76, but still lower than ours. The performance of the AlexNet-based model indicates that the IP component with the AlexNet architecture is less effective in extracting image features. Although the IP components using VGG16 and ResNet50 architectures show better performance than AlexNet, they still fall short of our method. Moreover, the results suggest that image-only models are less effective than text-only models, and both show lower performance compared to models developed by combining both types of data.

Although ResNet50, VGG16, and AlexNet are well-known CNN-based architectures that have significantly advanced the field of image processing and recognition, our model integrates the strengths of CNNs in capturing local patterns and the Transformer's prowess in modeling long-range dependencies. This hybrid model leverages the spatial

**Table 1 Refined dataset for model training and testing.**

| Data | Category | | | | | | |
|---|---|---|---|---|---|---|---|
| | Fires | Floods | Natural landscapes | Infrastructure | Human | Non-damage | Total |
| Training | 308 | 282 | 416 | 1,111 | 192 | 2,413 | 4,722 |
| Validation | 40 | 34 | 48 | 137 | 20 | 246 | 525 |
| Test | 36 | 30 | 50 | 142 | 28 | 297 | 583 |
| Total | 384 | 346 | 514 | 1,390 | 240 | 2,956 | 5,830 |

**Table 2 The performance of all the models.**

| Type data | Model | Accuracy | Precision | Recall | F1 score |
|---|---|---|---|---|---|
| Image | VGG16 | 0.7101 | 0.6988 | 0.5567 | 0.5467 |
| | ALexNet | 0.7479 | 0.6551 | 0.6406 | 0.6388 |
| | ResNet50 | 0.7530 | 0.6912 | 0.5941 | 0.5997 |
| Text | BERT | 0.7616 | 0.7214 | 0.6308 | 0.6649 |
| | XML | 0.7461 | 0.7135 | 0.5976 | 0.6197 |
| | RoBERTa | 0.7496 | 0.6606 | 0.6433 | 0.6438 |
| Image + Text | AlexNet + BERT | 0.7770 | 0.7225 | 0.7095 | 0.7119 |
| | VGG16 + BERT | 0.8216 | 0.8042 | 0.7408 | 0.7680 |
| | ResNet50 + BERT | 0.8216 | 0.8562 | 0.7090 | 0.7620 |
| | Our method | 0.8473 | 0.8530 | 0.7564 | 0.7947 |

hierarchies learned by CNNs and the global context captured by Transformers, offering a more holistic understanding of the data. Such a combination not only addresses the limitations of purely convolutional or purely attention-based methods but also leads to superior performance in tasks requiring nuanced understanding and integration of both local and global features, showcasing the significant advantage of the Convolutional-Transformer combinatory model over traditional CNN architectures like ResNet50, VGG16, and AlexNet.

## Statistical analysis

To provide statistical evidence on the stability of the proposed method, we repeated the experiment five times and recorded the performance of each trial. For each trial, the experiment was randomly resampled to create different training, validation, and test datasets. These datasets for model development and evaluation were independent of each other. Table 3 summarizes the results of the modeling experiment over five trials. The findings demonstrate that our proposed method is stable and reproducible with low variance over multiple trials with different sets of sampled data.

## Limitations

Besides preliminary achievements, multi-modal deep learning models like ours have several limitations. Properly aligning and synchronizing the data is challenging, and the

**Table 3 The performance of the proposed model over five different trials.**

| Trial | Accuracy | Precision | Recall | F1 score |
|---|---|---|---|---|
| 1 | 0.8473 | 0.8530 | 0.7564 | 0.7947 |
| 2 | 0.8542 | 0.8635 | 0.7553 | 0.7997 |
| 3 | 0.8611 | 0.8539 | 0.7719 | 0.8063 |
| 4 | 0.8508 | 0.8601 | 0.7751 | 0.8035 |
| 5 | 0.8370 | 0.8087 | 0.7658 | 0.7822 |
| Mean | 0.8501 | 0.8479 | 0.7649 | 0.7973 |
| Standard deviation | 0.0079 | 0.0200 | 0.0080 | 0.0085 |

increased model complexity may lead to extensive computational costs and elongated training times. Extracting meaningful features from both modalities is difficult, raising the risk of overfitting. Furthermore, interpreting these multi-modal models is harder than single-modal ones since the roles of texts and images in each sample vary. In certain cases, the text may contain more information while the image has less (*e.g.*, blurry, low-quality), and *vice versa*. Finally, understanding domain knowledge is highly essential to improving the training strategy and enhancing the model's generalizability.

## CONCLUSION

In our study, we proposed a robust method for detecting damage from social media posts, combining both textual and visual information to enrich the feature set. The image features capture specific regions of destruction, while text features provide insights into the severity or urgency of the cases. Our model is characterized by its IP component. Comparative analysis has shown that our IP component performs better than those developed with existing SOTA architectures. This method could be expanded in the future to integrate additional information (*e.g.*, human voice, ambient sounds) to further enhance performance. The blend of Transformer and convolutional layers in our approach captures both detailed and broad data aspects, providing a strong foundation for complex analysis. This synergy paves the way for advanced processing methods and innovations across various fields, with a critical understanding of the interplay between local and global data features essential for maximizing performance.

### Funding

The study was supported by the National Social Science Foundation of China (Project number: 22BXW016). The funders had no role in study design, data collection and analysis, decision to publish, or preparation of the manuscript.

### Grant Disclosures

The following grant information was disclosed by the authors:
National Social Science Foundation of China: 22BXW016.

## Competing Interests

The authors declare that they have no competing interests.

## Author Contributions

- Jiale Zhang conceived and designed the experiments, performed the experiments, analyzed the data, performed the computation work, prepared figures and/or tables, authored or reviewed drafts of the article, and approved the final draft.
- Manyu Liao conceived and designed the experiments, performed the experiments, analyzed the data, performed the computation work, prepared figures and/or tables, authored or reviewed drafts of the article, and approved the final draft.
- Yanping Wang conceived and designed the experiments, performed the experiments, analyzed the data, performed the computation work, prepared figures and/or tables, authored or reviewed drafts of the article, and approved the final draft.
- Yifan Huang conceived and designed the experiments, analyzed the data, authored or reviewed drafts of the article, and approved the final draft.
- Fuyu Chen conceived and designed the experiments, analyzed the data, authored or reviewed drafts of the article, and approved the final draft.
- Chiba Makiko conceived and designed the experiments, analyzed the data, authored or reviewed drafts of the article, and approved the final draft.

## Data Availability

The data used in this study is available at the University of Irvine:

Mouzannar,Hussein, Rizk,Yara, and Awad,Mariette. (2018). Multimodal Damage Identification for Humanitarian Computing. UCI Machine Learning Repository. https://doi.org/10.24432/C52P6P.

The Python code is available in the Supplemental File.

## Supplemental Information

Supplemental information for this article can be found online at http://dx.doi.org/10.7717/peerj-cs.2262#supplemental-information.

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
