# Peer review of "Multi-modal deep learning framework for damage detection in social media posts"

_PeerJ Computer Science, doi:10.7717/peerj-cs.2262_

## Round 0.1 · original submission · Major Revisions

Please address all the comments from the reviewers and revise the manuscript accordingly. Especially, consider including more methods in the experiments for comparison.

Reviewer 1 ·

Basic reporting

In this paper, authors proposed a novel approach for rich-textual and visual damage detection in social media platform through the integration between CNN-based and transformer (BERT)-based architectures. Generally speaking, the proposed ideas within this paper are considered as interesting in which the visual and textual information of social media’s posts have been combined within the data representation process in order to effectively improve the accuracy performance of the damage classification-driven fine-tuning process. Specifically, within the proposed technique in this paper, for the textual data the pre-trained BERT model is applied to learn the rich semantic/contextual feature representations. On the other hands, for the visual data (e.g. images) the integrated CNN/transformer-based blocks are utilized to extract the visual latent representations in forms of feature maps which are later combined with the textual representations previously produced by BERT model to form the unified/rich textual-visual representations of input social network’s posts. Therefore, it can assist to significantly improve the accuracy performance of the damage detection problem which is handled within a linear task-driven layer in the end. To validate the effectiveness/outperformance of their proposed technique, authors also presented the extensive empirical studies within a real-world/benchmark social media dataset. The experimental results have indicated the superiority of their proposed technique in comparing with previous deep learning-based baselines. In general, the methodology as well as structure of this paper are also sufficiently clear and systematic to clarify for authors’ important contributions in this paper. Beside good points of this paper, I also have some questions/recommendations for authors to improve their paper’s quality, including:
1) First of all, for the contents within introduction, authors should revise the contents in order to make it more concentrating on the novel/original contributions of their works, which make them different from previous approaches and why they are considered as important for the visual-textual enhanced damage detection task?
2) Related to discussions within the related work section, more discussions about the advantages/disadvantages of previous techniques in order to provide a bird’s eye views on recent approaches of damage detection problem. Additionally, please also specify more on this section which are previous techniques’ drawbacks are considered as main authors’ research motivations in this paper.
3) There are some mathematical symbols/notations which are utilized within the methodological section are utilized without first-time explanations. Thus, I suggest authors to carefully check all notations which are utilized within all equations of this section and add missing descriptions for these notation. Moreover, for all vector/matrix-based variables/notations please also specify the dimensionality for these variables.
4) For the comparative method setup, please adds more descriptions on how previous comparative baselines (e.g., AlexNet, VGG, etc.) are set up for experiments in this paper.
5) As most of the comparative baselines are the visual-based approach, so for the text-based approach, might be some comparative techniques (e.g., transformer-based techniques, like as XLM, RoBERTa, etc.) can be added to make authors’ empirical studies more sufficient.
6) If possible, please extend the comparative studies within the empirical study section with recent state-of-the-art deep learning based classification techniques which can be applied for the visual/textual-based damage categorization problem.

Experimental design

Please refer to my "Basic reporting" section.

Validity of the findings

Please refer to my "Basic reporting" section.

Additional comments

No comment.

Cite this review as

Reviewer 2 ·

Basic reporting

The manuscript was prepared with professional English. The "Background" and "Related works" cover adequate information for both expert and novice readers. The structure of the manuscript is straight-forward and well-indicative. The research questions are clear. This work is fairly qualified and meets most basic criteria to be considered to be published in this journal. The idea is interesting but additional works need to be done to add more values to this work.

Experimental design

The proposed method is based on two types of inputs and the model architecture is built on Convolutional and Transformer Networks. Authors combined two types of input with an expectation to enhance the model performance.

Authors compared their work to transfer learning-based methods using AlexNet, VGG16, and ResNet50. I suggest that authors should compare the proposed method with other models developed with convolutional neural network and transformer network as baseline models to obtain more evidence indicating the combination help to improve the model performance.

Validity of the findings

According to journal's criteria, authors need to provide statistical evidence (e.g., mean, standard deviation) to support the achieved findings. Repeating experiments using different random trials is required to investigate the variation in model performance across multiple trials.

Cite this review as

---

## Round 0.2 · accepted · Accept

The authors have made major revisions to the manuscript and addressed all of the reviewers' comments. The manuscript is now ready for publication.

Reviewer 1 ·

Basic reporting

After reviewing all revision notes as well as responses of authors for mine as well as other reviewer’s – I confirmed that all revisions as well as questions from reviewers have been sufficiently fulfilled by authors. Thus, I thought this paper can be accepted for publication in this form. Thank.

Experimental design

Please refer to my "Basic reporting" section.

Validity of the findings

Please refer to my "Basic reporting" section.

Additional comments

Please refer to my "Basic reporting" section.

Cite this review as

Reviewer 2 ·

Basic reporting

The revised manuscript meet all the required basic criteria with comprehensible language and well-organized.

Experimental design

Experimental design is well explained. All experiments were fairly conducted with clear explanations.

Validity of the findings

The statistical analysis provide sufficient evidence to convince readers about the robustness and producibility of this work.

Additional comments

I'm happy with the revised version. All of my concerns have been addressed.

Cite this review as